# Synthesis and Antiparasitic Activity of New Conjugates—Organic Drugs Tethered to Trithiolato-Bridged Dinuclear Ruthenium(II)–Arene Complexes

Oksana Desiatkina [1], Serena K. Johns [1,2], Nicoleta Anghel [3], Ghalia Boubaker [3], Andrew Hemphill [3,*], Julien Furrer [1,*] and Emilia Păunescu [1,*]

[1] Department of Chemistry, Biochemistry and Pharmaceutical Sciences, University of Bern, Freiestrasse 3, 3012 Bern, Switzerland; oksana.desiatkina@dcb.unibe.ch (O.D.); johnsserena.sj@gmail.com (S.K.J.)

[2] School of Chemistry, Cardiff University, Park Place, Cardiff CF10 3AT, UK

[3] Vetsuisse Faculty, Institute of Parasitology, University of Bern, Länggass-Strasse 122, 3012 Bern, Switzerland; nicoleta.anghel@vetsuisse.unibe.ch (N.A.); ghalia.boubaker@vetsuisse.unibe.ch (G.B.)

* Correspondence: andrew.hemphill@vetsuisse.unibe.ch (A.H.); julien.furrer@dcb.unibe.ch (J.F.); paunescu_emilia@yahoo.com (E.P.); Tel.: +41-31-6842384 (A.H.); +41-31-6844383 (J.F.); Fax: +41-31-6312477 (A.H.)

**Abstract:** Tethering known drugs to a metalorganic moiety is an efficient approach for modulating the anticancer, antibacterial, and antiparasitic activity of organometallic complexes. This study focused on the synthesis and evaluation of new dinuclear ruthenium(II)–arene compounds linked to several antimicrobial compounds such as dapsone, sulfamethoxazole, sulfadiazine, sulfadoxine, triclosan, metronidazole, ciprofloxacin, as well as menadione (a 1,4-naphtoquinone derivative). In a primary screen, 30 compounds (17 hybrid molecules, diruthenium intermediates, and antimicrobials) were assessed for in vitro activity against transgenic *T. gondii* tachyzoites constitutively expressing β-galactosidase (*T. gondii* β-gal) at 0.1 and 1 μM. In parallel, the cytotoxicity in noninfected host cells (human foreskin fibroblasts, HFF) was determined by an alamarBlue assay. When assessed at 1 μM, five compounds strongly impaired parasite proliferation by >90%, and HFF viability was retained at 50% or more, and they were further subjected to *T. gondii* β-gal dose-response studies. Two compounds, notably **11** and **13**, amide and ester conjugates with sulfadoxine and metronidazole, exhibited low IC$_{50}$ (half-maximal inhibitory concentration) values 0.063 and 0.152 μM, and low or intermediate impairment of HFF viability at 2.5 μM (83 and 64%). The nature of the anchored drug as well as that of the linking unit impacted the biological activity.

**Keywords:** bioorganometallic; trithiolato-bridged dinuclear ruthenium (II)–arene complexes; antiparasitic compounds; conjugates; *Toxoplasma gondii*

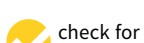

## 1. Introduction

In the last two decades, important research in the fight against cancer was focused on the use of ruthenium compounds as alternatives to platinum drugs currently employed as therapeutics [1–4]. Among the prominent compounds which have opened the way for recent research in this field remain ruthenium(III) complexes **NKP-1339** (sodium *trans*-[tetrachloridobis(1*H*-indazole)ruthenate(III)]) [5–7] and **NAMI-A** (imidazolium [*trans*-[tetrachlorido(*S*-dimethylsulfoxide)-(1*H*-imidazole)ruthenate(III)]) [6,8], as well as ruthenium(II)–arene complexes **RAPTA-C** ([Ru(II)($\eta^6$-*p*-MeC$_6$H$_4$Pr$^i$)Cl$_2$(PTA)], PTA = 1,3,5-triaza-7-phosphoadamantane) [9–11] and **RM175** ([Ru(II)($\eta^6$-biphenyl)Cl(en)]PF$_6$, en = 1,2-ethylenediamine) [7,12,13]. The ruthenium(II)–arene organometallic scaffold proved to be a versatile platform for the design of novel bioorganometallic agents and cationic or neutral compounds with the general structure [($\eta^6$-arene)Ru(X)(Y)(Z)] presenting both hydrophobic and hydrophilic properties have attracted increasing attention [14]. Following the work pioneered by **RAPTA-C** and **RM175**, studies on organometal-

lic ruthenium(II)–arene complexes are not only rapidly progressing but also particularly relevant, with a plethora of compounds presenting anticancer, antibiotic, antifungal and antiparasitic properties [15–20]. Important research focused on hybrid structures in which the robust ruthenium(II)–arene unit is associated with the various biologically active compounds, a strategy with implications for developing of novel metal-based compounds presenting multiple targets [21].

Trithiolato-bridged dinuclear ruthenium(II)–arene complexes constitute a particular family of ruthenium(II)–arene compounds, whose structure is based on two half-sandwich units linked by three thiols forming a trigonal-bipyramidal unit (Figure 1). Two types of structures can be distinguished, namely symmetric complexes (Figure 1, **A** and **A'**) in which the three thiols are identical (general formula [($\eta^6$-$p$-MeC$_6$H$_4$Pr$^i$)$_2$Ru$_2$($\mu_2$-SR)$_3$]X), and mixed complexes (Figure 1, **B**) bearing at least one different thiol (general formula [($\eta^6$-$p$-MeC$_6$H$_4$Pr$^i$)$_2$Ru$_2$($\mu_2$-SR1)$_2$($\mu_2$-SR2)]X). These compounds have been initially developed and evaluated as catalysts [22], and subsequently as cytotoxic agents. Particularly complex **A'** (R = Bu$^t$) was highly active against in vitro cultured cancer cells [23–27], and three analogues were also tested in vivo [28,29]. This encouraged us to assess trithiolato diruthenium complexes as potential antiparasitic agents, and several derivatives were highly active against *Toxoplasma gondii* [30], *Neospora caninum* [31] and *Trypanosona brucei* [32]. The half-maximal proliferation inhibitory concentrations (IC$_{50}$) of complexes **A** (R = Me), **A'** (R = Bu$^t$) and **B** against in vitro cultured *T. gondii* tachyzoites were 34, 62 and 1 nM, respectively, and these compounds did not impair the viability of human foreskin fibroblast (HFF) host cells [30].

**Figure 1.** Structure of selected symmetric (**A**, **A'**) and mixed (**B**) trithiolato-bridged ruthenium(II)–arene complexes and of the antimicrobial drugs considered for tethering to the diruthenium unit.

Toxoplasmosis is considered one of the most common parasitic diseases affecting approximately one-third of the world's population. In immunocompetent hosts, the infection is usually controlled and asymptomatic, but in immunocompromised persons, such as AIDS patients or persons undergoing immunosuppressive therapy, newly acquired reactivated toxoplasmosis can cause serious complications such as toxoplasmic encephalitis or ocular toxoplasmosis [33], and primary infection during pregnancy, can lead to abortions or fetal malformation. The current therapeutic options are suboptimal, target only the acute disease, and do not eradicate the parasite in chronic infections encysted organisms (bradyzoites) [34,35]. Additionally, adverse side effects are frequently reported [34,36]. Thus, safer, and more effective treatment options are needed.

Cationic trithiolato dinuclear ruthenium(II)–arene complexes represent promising scaffolds. Tethering a functional molecule (e.g., fluorophore, metabolite, drug) to a metal framework is one of the strategies used for tracking, directing, or modulating the biological activity of metal-based complexes [37–39]. In numerous cases, conjugation to organometallic moieties led to the enhanced biological activity of the parental drug [4,40]. The trithiolato diruthenium compounds showed high stability and post-functionalization of the bridge thiols proved a useful method to introduce valuable modifications [41–43]. This approach proceeds under mild conditions and constitutes an efficient entry to conjugates with biorelevant moieties. The easy access and upscaling of certain mixed trithiolato diruthenium compounds bearing derivatizable groups (e.g., OH, SH, $NH_2$, $CO_2H$) [27,43] (as **B**, Figure 1) allowed the development and investigation of hybrid structures functionalized with the anticancer drug chlorambucil [41], 7-amino-coumarin fluorophores [43], and peptides [42]. This type of structure is aimed at improving the cytotoxicity against cancer cells [41,42], the water solubility [42] and the selectivity and antiparasitic activity on *T. gondii* [43].

In this study, a series of structurally diverse antimicrobial agents presenting also relevant anti-*Toxoplasma* activity have been selected to be conjugated to the trithiolato diruthenium core (Figure 1) [34,36,44,45]. Ruthenium(II)–arene, as well as iridium(III)- and rhodium(III)-cyclopentadienyl are among the favored organometallic units to be tested in intramolecular combination with various drugs.

Sulfonamides (sulfa-drugs) such as sulfadiazine (4-amino-*N*-(pyrimidin-2-yl) benzenesulfonamide), sulfamethoxazole (4-amino-*N*-(5-methylisoxazol-3-yl)benzenesulfonamide), and sulfadoxine (4-amino-*N*-(5,6-dimethoxypyrimidin-4-yl)benzenesulfonamide) (Figure 1) are bacteriostatic agents presenting a broad-spectrum of activity against Gram-positive and Gram-negative bacteria, *Toxoplasma* and other protozoan pathogens. Drug combinations including pyrimethamine/sulfadiazine and trimethoprim/sulfamethoxazole remain among the most frequently applied treatments for toxoplasmosis [34,36,46–48]. Several studies focused on combining sulfa-drugs with various half-sandwich 'piano-stool' organometallic units for biological applications [49–51] (e.g., **C** and **D** in Figure 2). The sulfa-drug can be directly coordinated to the organometallic unit (**C**) [49], or can be covalently connected to a ligand (**D**) [50,51]. 'Piano-stool' half-sandwich compounds of the type $[(\eta^6\text{-}p\text{-MeC}_6\text{H}_4\text{Pr}^i)\text{Ru(sulfadiazine)}_2]$ and $[(\eta^5\text{-C}_5\text{Me}_5)\text{Rh(sulfadiazine)}_2]$ (where $\text{C}_5\text{Me}_5$ is pentamethylcyclopentadienyl) (**C**, Figure 2) were evaluated for their potential antimicrobial activity [49]. While the ruthenium complex was biologically inactive, the rhodium compound was potent against Gram-positive bacteria, *Candida albicans* and *Cryptococcus neoformans* [49].

Another example is a series of organo-ruthenium, rhodium and iridium derivatives of sulfadoxine anchored on *N*,*N*′-chelate pyridylimino quinolylimino-bidentate ligands [50] (e.g., **D**, Figure 2). Screening for in vitro activity against *Plasmodium falciparum* chloroquine-sensitive and -resistant strains and *Mycobacterium tuberculosis* showed the activity to be dependent on the organometallic unit, with ruthenium complexes being inactive. The rhodium and iridium compounds inhibited parasite growth with $IC_{50}$ values in the sub- and low micromolar range, with no significant toxicity towards human embryonic kidney cells (HEK293). Moreover, sulfadoxine was not active in most of the assays, supporting the hypothesis that organometallic conjugates of drugs can beneficially affect bioactivity.

Triclosan (5-chloro-2-(2,4-dichlorophenoxy)phenol, Figure 1) is an antibacterial agent that was shown to inhibit the in vitro proliferation of *T. gondii* tachyzoites in the low nanomolar range [52,53]. As triclosan presents poor water solubility and oral bioavailability, various studies focused on derivatives aiming at increased solubility and potency [54–56], as well as improved drug delivery and pharmacological properties against *T. gondii* [57–59].

**Figure 2.** Structure of half-sandwich RhCp* sulfadiazine complex (**C**), IrCp*biph sulfadoxine conjugate **D**, ruthenium(II)–arene conjugates with metronidazole (**E**), ciprofloxacin (**F** and **G**), lapachol (**H**), and plumbagin (**I**).

Metronidazole (1-β-hydroxyethyl-2-methyl-5-nitroimidazole, Figure 1) is used to treat antibacterial and antiprotozoal infections [60,61]. A significant reduction of brain cysts was observed in a mouse model of chronic toxoplasmosis after combined treatment with spiramycin and metronidazole [62]. The potential use of ruthenium(II)–benzene metronidazole complex **E** as a hypoxic cell cytotoxic agent has been assessed, revealing a higher selective toxicity for **E** compared to the free metronidazole [63].

Fluoroquinolones antibiotics (such as ciprofloxacin (1-cyclopropyl-6-fluoro-4-oxo-7-(piperazin-1-yl)-1,4-dihydroquinoline-3-carboxylic acid), Figure 1) are widely used in human and veterinary medicine. Along with other fluoroquinolones, previous reports identified promising anti-*Toxoplasma* effects upon treatments with certain ciprofloxacin derivatives [64,65]. Ruthenium(II)–arene complexes bearing fluoroquinolone-based ligands showed interesting anticancer and antimicrobial activities (e.g., **F** and **G** in Figure 2) [66–68]. The drug can be either directly coordinated to the metal (**F**), or covalently linked to a ligand (**G**). Coordinating 7-(4-(decanoyl)piperazin-1-yl)-ciprofloxacin, to the ruthenium(II)($\eta^6$-$p$-MeC$_6$H$_4$Pr$^i$) unit in compound **F** [67] yielded multifunctional properties: high cytotoxicity in several cancer cell lines and moderate dose-dependent antibacterial activity in *Escherichia coli* as well as in a clinical *E. coli* isolate resistant to β-lactams. Compound **G**, bearing ciprofloxacin connected to an aminomethyl(diphenyl)phosphine ligand, was loaded in polymeric nanoformulations and exhibited promising cytotoxicity in vitro [66].

Derivatives containing the 1,4-naphtoquinone moiety, including atovaquone (2-((1$r$,4$r$)-4-(4-chlorophenyl)cyclohexyl)-3-hydroxynaphthalene-1,4-dione, Figure 1), buparvaquone (2-((4-(tert-butyl)cyclohexyl)methyl)-3-hydroxynaphthalene-1,4-dione) or lapachol (2-hydroxy-3-(3-methylbut-2-en-1-yl)naphthalene-1,4-dione) exhibit interesting anticancer and antiparasitic properties [69–73]. Atovaquone (Figure 1) is active in vitro against *T. gondii* tachyzoites with low nanomolar IC$_{50}$ values and is one of the drugs currently applied to treat acute toxoplasmosis in humans [34,36,48]. Buparvaquone is also highly active against *T. gondii*

in vitro and was shown to limit cerebral infection of dams and vertical transmission in mice infected with *T. gondii* oocysts [70]. Ruthenium(II)–arene complexes with naphtoquinone-based ligands have shown potential anticancer properties [74–78] (e.g., **H** and **I** in Figure 2). Ruthenium(II)–*p*-MeC$_6$H$_4$Pr$^i$ complex **H** with lapachol as a bidentate ligand was shown to induce apoptosis in human cancer cells in the low micromolar range by a mode of action involving oxidative stress [77]. Hydrazone-linked plumbagin ruthenium(II) conjugate I showed distinct and selective cancer cell growth inhibition, stronger DNA binding than plumbagin, and Pgp (P-glycoprotein) transporter inhibition [74].

This study aimed to synthesize new conjugates trithiolato-bridged binuclear ruthenium(II)–arene unit antimicrobial drug. The library of active organic compounds comprised sulfa-drugs (dapsone, sulfamethoxazole, sulfadiazine, sulfadoxine), triclosan, metronidazole, ciprofloxacin and menadione (2-methylnaphthalene-1,4-dione) (Figure 1). In addition to the nature of the antimicrobial drug, different structural variations were investigated for the hybrid molecules, as the connector between the two components (ester, amide, triazole) and the relative proportion drug unit/diruthenium moiety (i.e., 1:1 vs. 3:1).

The antimicrobial drugs, the newly obtained conjugates and the associated intermediates were submitted to a first in vitro screening, assessing the activity against a transgenic *T. gondii* strain constitutively expressing β-galactosidase (*T. gondii* β-gal) grown in human foreskin fibroblasts (HFF). In parallel, the cytotoxicity of these compounds was evaluated in noninfected HFF by the alamarBlue assay. The compounds exhibiting interesting antiparasitic activity and low cytotoxicity were subjected to *T. gondii* IC$_{50}$ determination.

## 2. Results and Discussion

### 2.1. Chemistry

#### 2.1.1. Synthesis of the Trithiolato-Bridged Diruthenium Intermediates

To access the hybrid molecules, selected diruthenium intermediates with groups allowing further modification via ester/amide conjugation or click chemistry (CuAAC, Cu(I)-catalyzed azide-alkyne cycloaddition) were synthesized (Schemes 1 and 2). The dithiolato ruthenium precursor **1** was synthesized by the reaction of the ruthenium dimer ([Ru($\eta^6$-*p*-MeC$_6$H$_4$Pr$^i$)Cl$_2$]$_2$) with two equivalents of 4-(*tert*-butyl)phenyl)methanethiol (Scheme 1) as previously reported [43].

**Scheme 1.** Synthesis of the dithiolato diruthenium precursor **1**, of the mixed trithiolato intermediates **2**, **3**, and **4**, and the symmetric trithiolato intermediates **5** and **6**.

**Scheme 2.** Synthesis of the diruthenium trithiolato alkyne intermediate **7**.

**1** was further reacted with appropriate thiols (2-(4-mercaptophenyl)acetic acid, 4-mercaptophenol and 4-aminobenzenethiol), leading to the obtainment of mixed trithiolato diruthenium(II)–arene complexes **2**, **3**, and **4**, bearing carboxy [43], hydroxy [27,43] and, respectively, amino [43] groups (Scheme 1) following reported procedures.

To vary the number of drug units tethered to the diruthenium moiety, symmetric trithiolato intermediates **5** [24,43,79] and **6** [80] bearing three hydroxy or amino groups were also synthesized following reported protocols by reacting the ruthenium dimer ([Ru($\eta^6$-$p$-MeC$_6$H$_4$Pr$^i$)Cl]$_2$Cl$_2$) with 4-mercaptophenol and, respectively, 4-aminobenzenethiol in excess (Scheme 1).

In the conjugates, the nature of the linker between the diruthenium unit and the drug molecule might be very important for the stability and the biological activity of the hybrid molecule. To extend the purpose of this study, in addition to ester and amide bonds, the use of the triazole ring as a linker was also investigated.

Compound **7**, a diruthenium intermediate bearing a pending alkyne group, was synthesized in good yield (83%) by the amide coupling of carboxy derivative **2** and propargyl amine by adapting a reported procedure [43] (Scheme 2). HOBt (1-hydroxybenzotriazol) and EDCI (*N*-(3-dimethylaminopropyl)-*N*′-ethylcarbodiimide hydrochloride) were used as coupling agents in basic conditions (DIPEA, *N*,*N*-diisopropylethylamine).

### 2.1.2. Conjugates with Sulfa-Drugs (Dapsone, Sulfamethoxazole, Sulfadiazine, Sulfadoxine)

Conjugates **8**, **9**, **10** and **11** were obtained in modest yields (29, 24, 24 and 35%, respectively) by the amide coupling of the carboxy diruthenium intermediate **2** with commercially available sulfa-drugs dapsone, sulfamethoxazole, sulfadiazine and sulfadoxine in the presence of the coupling agents HOBt and EDCI, in basic conditions (DIPEA) (Scheme 3). The reduced solubility of the starting amines led to poor conversions and yields of isolated pure compounds.

### 2.1.3. Conjugates with Triclosan and Metronidazole

Ester conjugates with triclosan and metronidazole **12** and **13** were obtained by reacting carboxy complex **2** with the corresponding drugs, both containing free hydroxy groups (Scheme 3). Reactions were performed using EDCI as a coupling agent and DMAP (4-(dimethylamino)pyridine) as a basic catalyst, compounds **12** and **13** being isolated in medium yields of 40 and 51%.

The 'click' metronidazole conjugate **15** was synthesized by the 1,3-dipolar cycloaddition reaction of the alkyne diruthenium intermediate **7** with the metronidazole azide derivative **14** performed in the presence of CuSO$_4$ as a catalyst and sodium ascorbate as a reducing agent (Scheme 4), using an adapted literature procedure [81,82]; conjugate **15** was isolated in 33% yield. The metronidazole azide **14** was synthesized in two steps (activation of the hydroxy group as mesylate followed by the nucleophilic substitution with azide). Of note, this is the first time that the 1,3-dipolar cycloaddition reaction is used for synthesizing conjugates based on the trithiolato diruthenium scaffold.

**Scheme 3.** Synthesis of the amide conjugates with the sulfa-drugs dapsone **8**, sulfamethoxazole **9**, sulfadiazine **10,** sulfadoxine **11**, and ester conjugates with triclosan **12** and metronidazole **13**.

**Scheme 4.** Synthesis of metronidazole 'click' conjugate **15**.

### 2.1.4. Conjugates with Ciprofloxacin

For the derivatization of ciprofloxacin, two positions can be considered: the carboxy group in position 3 or the piperazine fragment in position 7 of the fluoroquinolone core [83]. To avoid possible side reactions, the protection of one of these groups was considered prior to attempt connecting this moiety to the trithiolato diruthenium unit. The piperazine fragment of ciprofloxacin was protected using Boc₂O (di-*tert*-butyl dicarbonate) in

basic conditions (TEA, triethylamine) following a reported protocol [84] (Scheme 5), and intermediate **16** was isolated in quantitative yield.

**Scheme 5.** Synthesis of the ciprofloxacin conjugates **17**, **18** and **19**.

The 'mixed' hydroxy and amino diruthenium complexes **3** and **4** were reacted with piperazine *N*-Boc protected ciprofloxacin **16** (Scheme 5). The esterification (conjugate **17**) was realized in the presence of EDCI and DMAP, while the amide coupling (conjugate **18**) was performed in the presence of HOBt, EDCI, and DIPEA. While the amide conjugate **18** was easily isolated in high yield (87%), the ester conjugate **17** could not be obtained in pure form, as it is prone to hydrolysis/solvolysis during purification. The *N*-Boc deprotection of **18** was realized in classical acidic conditions [85,86] (TFA, trifluoroacetic acid, Scheme 5), allowing the isolation of compound **19** in 64% yield.

### 2.1.5. Conjugates with Menadione

Since atovaquone (Figure 1) and buparvaquone are quinone-based antimicrobial medications for the prevention and treatment of *T. gondii* [69,70,72] and other parasites, we have considered the development of a small library of compounds in which the 1,4-naphtoquinone motif is associated with the trithiolato diruthenium scaffold. To validate the concept, a simpler structure based on the menadione moiety (2-methylnaphthalene-1,4-dione, Figure 1) was approached. First, menadione carboxy derivatives **20**, **21** and **22** that can be further anchored on the diruthenium unit were synthesized. This type of modification was previously used to prepare carboxy analogues of lawsone (2-hydroxynaphthalene-1,4-dione) [87–89], menadione or plumbagin (5-hydroxy-2-methylnaphthalene-1,4-dione) [90–93]. Compounds **20**, **21** and **22**, bearing linkers of different lengths between the 1,4-naphtoquinone moiety and the carboxylic group, were obtained from menadione and succinic, suberic and adipic acid, respectively, in the presence of $AgNO_3$ and $(NH_4)_2S_2O_3$ following literature procedures [90], and were isolated in medium yields of 70, 61 and 52%, respectively (Scheme 6).

**Scheme 6.** Synthesis of carboxylic acid-functionalized 1,4-naphtoquinone derivatives **20**, **21** and **22**.

**20** was further reacted with the hydroxy and amino diruthenium complexes **3** and **4** (Scheme 7). The esterification reaction was realized in the presence of EDCI and DMAP and important issues were encountered in the purification of conjugate **23** due to degradation (59%). The amide coupling was performed in the presence of HOBt, EDCI and DIPEA, and led to the isolation of **24** in medium yield (54%).

**Scheme 7.** Synthesis of 1,4-naphtoquinone ester and amide conjugates **23** and **24**.

Varying the relative proportion between the metal units and the drug fragments in this type of conjugates might lead to improved bio-efficacy [94,95]. To increase the number of 1,4-naphtoquinone molecules anchored on the trithiolato diruthenium core, symmetric intermediates **5** and **6** bearing either three hydroxy or three amino groups were used for the ester and amide couplings with **20**, **21** and **22** (Scheme 8). The reaction of **5** with the carboxy 1,4-naphtoquinone derivative **20** in the presence of HOBt, EDCI and DIPEA, allowed only the isolation of the monosubstituted conjugate **25** in a low yield of 27%. At the same time, important degradation of the diruthenium substrate was observed.

**Scheme 8.** Synthesis of 1,4-naphtoquinone conjugates **25**, **26**, **27** and **28**.

The reactions of **6** with the menadione carboxy derivatives **20** and **21** in the presence of HOBt, EDCI, and DIPEA led to the isolation of the tri-amide conjugates **26** and **27** with low yields of 8% and 33%, respectively (Scheme 8). However, the reaction of **6**

with the 1,4-naphtoquinone analogue **22** performed in similar conditions led only to the monosubstituted conjugate **28**, isolated with a medium yield of 50%.

All compounds were analyzed and characterized by [1]H, [13]C and [19]F (where suitable) nuclear magnetic resonance spectroscopy (NMR), electrospray ionization mass spectrometry (ESI-MS) and elemental analysis (see Experimental part-Chemistry in the Supplementary Materials for full details). ESI-MS corroborated the spectroscopic data with the dithiolato precursor **1**, the trithiolato diruthenium intermediates **2–7** and the conjugates **8–11** (sulfa-drugs), **12** (triclosan), **13** and **15** (metronidazole), **17–19** (ciprofloxacin) and **23–28** (menadione) exhibiting molecular ion peaks corresponding to [M-Cl]$^+$ ions.

For the assessment of the biological activity, the compounds were prepared as stock solutions in dimethylsulfoxide (DMSO), in which the compounds are well soluble. [1]H-NMR spectra of similar conjugates (with polypeptides, coumarin units or derivatives with two or three diruthenium units) dissolved in DMSO-$d_6$ or deuterated water, recorded at 25 °C 5 min and 28 days after sample preparation showed no visible changes, demonstrating very good stability of the compounds in this highly complexing solvent and in water [42,43,80,96].

### 2.1.6. X-ray Crystallography

The crystal structure of the trithiolato diruthenium sulfamethoxazole conjugate **9** was established in the solid state by single-crystal X-ray diffraction (an ORTEP representation is shown in Figure 3), confirming the expected structure. To the best of our knowledge, this is the first example of a structure containing the trithiolato-bridged diruthenium unit and an organic moiety. Data collection and refinement parameters are given in Table 1. Selected bond lengths and angles are presented in Table 2.

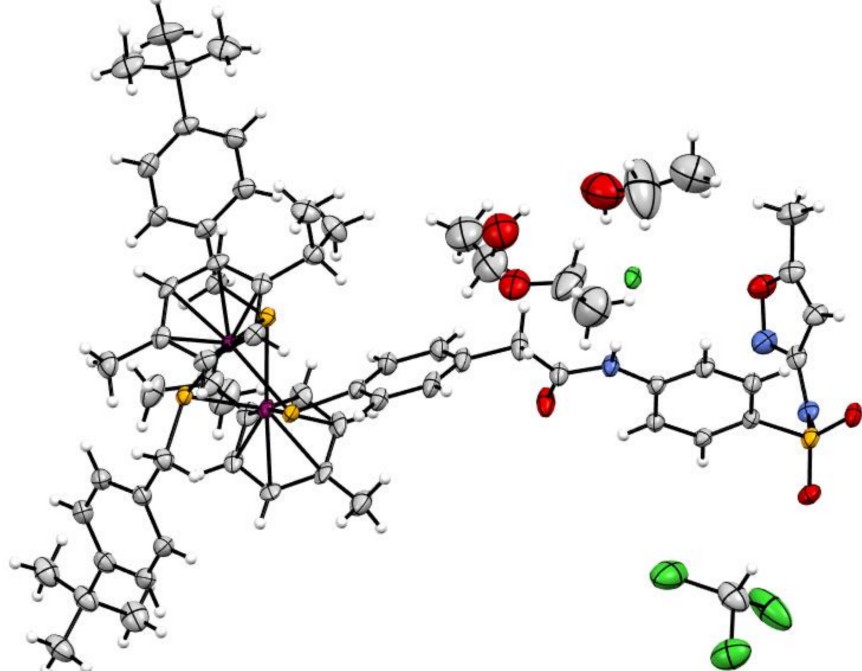

**Figure 3.** ORTEP representation of complex **9** (thermal ellipsoids are 50% equiprobability envelopes, and H atoms are spheres of arbitrary diameter; the asymmetric unit contains one organometallic complex, three EtOH, and one CHCl$_3$ molecule).

**Table 1.** Crystal data and structure refinement for **9**.

| Compound | 9 |
|---|---|
| Formula | $C_{60}H_{74}ClN_3O_4Ru_2S_4 \cdot 3CH_3CH_2OH \cdot CHCl_3$ |
| F.W. (g·mol$^{-1}$) | 1524.62 |
| Temperature (K) | 110.2(5) |
| Crystal system | Monoclinic |
| Space group | $P2_1/n$ |
| a (Å) | 14.13470(10) |
| b (Å) | 24.4657(2) |
| c (Å) | 20.7272(2) |
| $\alpha$ (°) | 90 |
| $\beta$ (°) | 100.0580(10) |
| $\gamma$ (°) | 90 |
| $V$ (Å$^3$) | 7057.62(10) |
| Z | 4 |
| $D_{calc}$ (g·cm$^{-3}$) | 1.435 |
| μ (mm$^{-1}$) | 6.38 |
| F(000) | 3168 |
| Crystal size (mm$^3$) | $0.2 \times 0.075 \times 0.05$ |
| Θ range for data collection (°) | 5.64 to 154.266 |
| Index ranges | |
| h | −17/12 |
| k | −30/30 |
| l | −26/25 |
| Reflns. collected | 56,233 |
| Independent reflns. | 14,528 [$R_{int}$ = 0.0442, $R_{sigma}$ = 0.0350] |
| Data/restraints/parameters | 14,528/2/803 |
| GoodF$^2$ | 1.047 |
| R1 [I ≥ 2σ(I)] | 0.0564 |
| wR2 | 0.1597 |
| R1 [all data] | 0.0613 |
| wR2 | 0.1645 |
| Largest diff. peak/hole (Å$^{-3}$) | 3.23/−1.47 |

**Table 2.** Comparison of key bond lengths (Å) and angles (°) of the diruthenium moiety in **9** and previously reported mixed complex **J** (Figure 4, data from ref. [97]).

| | Complex 9 | Complex J |
|---|---|---|
| Ru-S | Ru(1)-S(1) 2.3749(10) Ru(1)-S(2) 2.3927(10) Ru(1)-S(3) 2.3973(10) Ru(2)-S(1) 2.3884(10) Ru(2)-S(2) 2.3931(11) Ru(2)-S(3) 2.3869(10) | Ru(1)-S(1) 2.3878(9) Ru(1)-S(2) 2.4023(9) Ru(1)-S(3) 2.3813(8) Ru(2)-S(1) 2.3992(9) Ru(2)-S(2) 2.3991(8) Ru(2)-S(3) 2.3882(8) |
| Ru-$\eta6$ | Ru(1)-cent(C21-C26) Ru(2)-cent(C31-C36) | Ru(1)-cent(C1-C6) 1.708 Ru(2)-cent(C11-C16) 1.709 |
| S-Ru-S | S(1)-Ru(2)-S(2) 76.29(4) S(1)-Ru(2)-S(3) 74.94(4) S(2)-Ru(2)-S(3) 77.32(4) S(1)-Ru(1)-S(2) 76.56(3) S(1)-Ru(1)-S(3) 75.00(4) S(2)-Ru(1)-S(3) 77.13(4) | S(1)-Ru(1)-S(2) 74.95(3) S(1)-Ru(1)-S(3) 77.72(3) S(2)-Ru(1)-S(3) 75.75(3) S(1)-Ru(2)-S(2) 74.81(3) S(1)-Ru(2)-S(3) 77.37(3) S(2)-Ru(2)-S(3) 75.68(3) |
| Ru-S-Ru | Ru(1)-S(1)-Ru(2) 89.45(3) Ru(1)-S(2)-Ru(2) 88.91(3) Ru(1)-S(3)-Ru(2) 88.95(4) | Ru(1)-S(1)-Ru(2) 89.27(3) Ru(1)-S(2)-Ru(2) 88.93(3) Ru(1)-S(3)-Ru(2) 89.68(3) |

**Table 2.** *Cont.*

|  | Complex 9 | Complex J |
|---|---|---|
| Ru-cent(S-S-S)-Ru | Ru(1)-cent(S1-S3)-Ru(2)<br>178.71 | Ru(1)-cent(S1-S3)-Ru(2)<br>177.30 |
| cent η6-cent(S-S-S)-cent η6 | cent(C24-C29)-cent(S1-S3)-cent(C72-C77)<br>177.74 | cent(C1-C6)-cent(S1-S3)-cent(C11-C16)<br>176.25 |

Cent—represents the centroid calculated using Mercury CCDC 4.1.2.

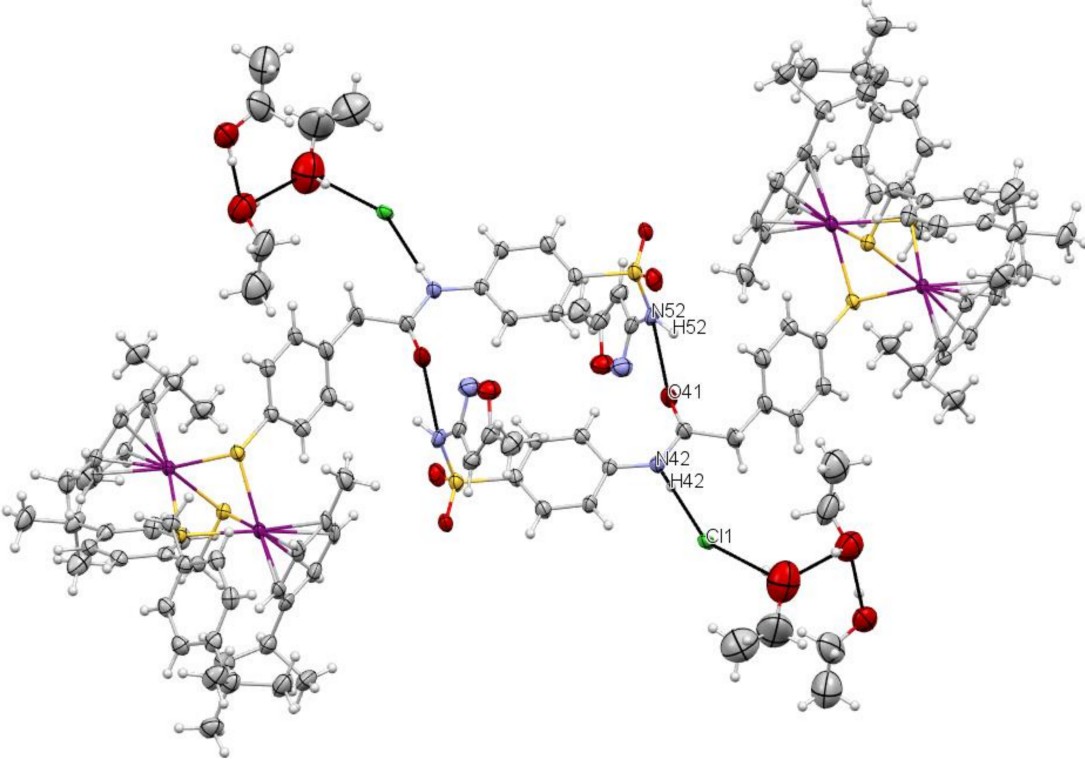

**Figure 4.** Structure of complex **J**, $[(\eta^6\text{-}p\text{-}MeC_6H_4Pr^i)_2Ru_2(\mu_2\text{-}SCH_2\text{-}C_6H_5)_2(\mu_2\text{-}SC_6H_4\text{-}p\text{-}OH)]BH_4$. Data from ref. [97].

In the network, an organization in dimers due to the presence of intermolecular H-bonding interactions between the sulfamethoxazole fragments was observed (Figure 5). These interactions involve sulfonamide NH from one molecule and the carboxyamide oxygen atom of another molecule. Additional H-bonding interactions are observed between the carboxyamide NH and the Cl⁻ counterion. Representative bond lengths and angles for these interactions are given in Table 3.

**Figure 5.** Intermolecular H-bonding interactions in the crystal of **9** with the formation of dimers; two H-bonds interconnect the carboxyamide C=O groups from the two diruthenium complexes to the sulfonamide NH from the other molecule. Supplementary H-bonding interactions were observed between the carboxyamide NH and the Cl⁻ counterion (contacts D-H···A correspond to N-H···Cl⁻, N-H···O, image produced using Mercury CCDC 4.1.2, see bond parameters in Table 3).

**Table 3.** Intramolecular H-bonding interactions for complex **9**.

| Compound | Contact D-H⋯A | Distance (Å) | | | Angle (°) |
|---|---|---|---|---|---|
| | | *D*-H | H⋯*A* | *D*⋯*A* | *D*-H⋯*A* |
| **9** | N52-H52 . . . O41 | 0.860 | 2.162 | 2.786 | 129.22 |
| | N42-H42 . . . Cl1 | 0.859 | 2.407 | 3.254 | 169.15 |

*2.2. Assessment of the In Vitro Activity against the Apicomplexan Parasite Toxoplasma gondii*

2.2.1. Primary Screening

The activity against *T. gondii* tachyzoites and HFF (human foreskin fibroblasts) host cells of the new conjugates (13 compounds), of the antimicrobial drugs (8 compounds) and the representative intermediates (9 compounds) has been investigated. The trithiolato intermediates **2–6** have been evaluated previously against *T. gondii* β-gal under similar conditions [30,43,80], and the corresponding values were introduced in Table 4 and Figure 6 for comparison. Of note, complex **5** exhibited no activity against the parasite [30] and was therefore not included in the discussion of the results. The purity of isolated ciprofloxacin and menadione ester conjugates **17**, **23** and **25** was not satisfactory and, therefore, these compounds were not evaluated.

**Table 4.** Cytotoxicity/efficacy screening of compounds in noninfected HFF cultures and *T. gondii* β-gal tachyzoites cultured in HFF. Tests were realized in triplicate. The values of the compounds selected for determination of IC$_{50}$ values against *T. gondii* β-gal are highlighted in bold.

| Compound | HFF Viability (%) | | *T. gondii* β-gal Growth (%) | |
|---|---|---|---|---|
| | 0.1 μM | 1 μM | 0.1 μM | 1 μM |
| *Ruthenium intermediates* | | | | |
| **2** [a] | 91 ± 4 | 73 ± 1 | 114 ± 2 | 110 ± 2 |
| **3** [a] | 76 ± 6 | 46 ± 6 | 66 ± 14 | 2 ± 0 |
| **4** [a] | 74 ± 2 | 48 ± 1 | 57 ± 1 | 2 ± 0 |
| **6** [a] | 97 ± 4 | 61 ± 6 | 115 ± 4 | 85 ± 5 |
| **7** | 71 ± 2 | 46 ± 6 | 52 ± 13 | 3 ± 1 |
| *Conjugates with sulfa-drugs* | | | | |
| Dapsone | 92 ± 4 | 103 ± 3 | 77 ± 4 | 42 ± 0 |
| **8** | 104 ± 1 | 91 ± 2 | 148 ± 2 | 36 ± 2 |
| Sulfamethoxazole | 93 ± 3 | 102 ± 5 | 78 ± 5 | 75 ± 7 |
| **9** | 90 ± 12 | 63 ± 7 | 83 ± 8 | 77 ± 3 |
| Sulfadiazine | 101 ± 2 | 33 ± 3 | 57 ± 5 | 70 ± 5 |
| **10** | 113 ± 1 | 93 ± 2 | 72 ± 3 | 0 ± 0 |
| Sulfadoxine | 97 ± 3 | 104 ± 0 | 111 ± 3 | 83 ± 2 |
| **11** | 100 ± 3 | 100 ± 8 | 116 ± 1 | 11 ± 1 |
| *Conjugates with triclosan and metronidazole* | | | | |
| Triclosan | 99 ± 1 | 97 ± 1 | 80 ± 2 | 71 ± 2 |
| **12** | 100 ± 2 | 103 ± 1 | 76 ± 6 | 66 ± 12 |
| Metronidazole | 101 ± 2 | 100 ± 1 | 115 ± 8 | 116 ± 6 |
| **13** | 115 ± 2 | 93 ± 1 | 101 ± 7 | 1 ± 0 |
| **14** | 98 ± 3 | 97 ± 2 | 115 ±7 | 92 ± 1 |
| **15** | 116 ± 1 | 99 ± 1 | 98 ± 2 | 1 ± 0 |
| *Conjugates with ciprofloxacin* | | | | |
| Ciprofloxacin | 101 ± 1 | 99 ± 0 | 82 ± 3 | 84 ± 3 |
| **18** | 92 ± 0 | 89 ± 0 | 94 ± 2 | 102 ± 1 |
| **19** | 102 ± 2 | 94 ± 2 | 68 ± 4 | 21 ± 3 |
| *Conjugates with menadione* | | | | |
| Menadione | 117 ± 3 | 101 ± 4 | 103 ± 7 | 50 ± 2 |
| **20** | 105 ± 3 | 94 ± 2 | 101 ± 8 | 107 ± 3 |
| **21** | 109 ± 2 | 95 ± 1 | 86 ± 10 | 84 ± 5 |
| **22** | 110 ± 3 | 87 ± 1 | 83 ± 4 | 92 ± 1 |
| **24** | 95 ± 1 | 92 ± 2 | 65 ± 4 | 3 ± 0 |
| **26** | 101 ± 2 | 102 ± 1 | 89 ± 16 | 90 ± 7 |
| **27** | 100 ± 2 | 100 ± 3 | 164 ± 4 | 92 ± 3 |
| **28** | 98 ± 3 | 92 ± 2 | 71 ± 6 | 46 ± 1 |

[a] Data for compounds **2–6** were previously reported [30,43,80]. Complex **5** exhibited no activity against the parasite [30] (values not shown).

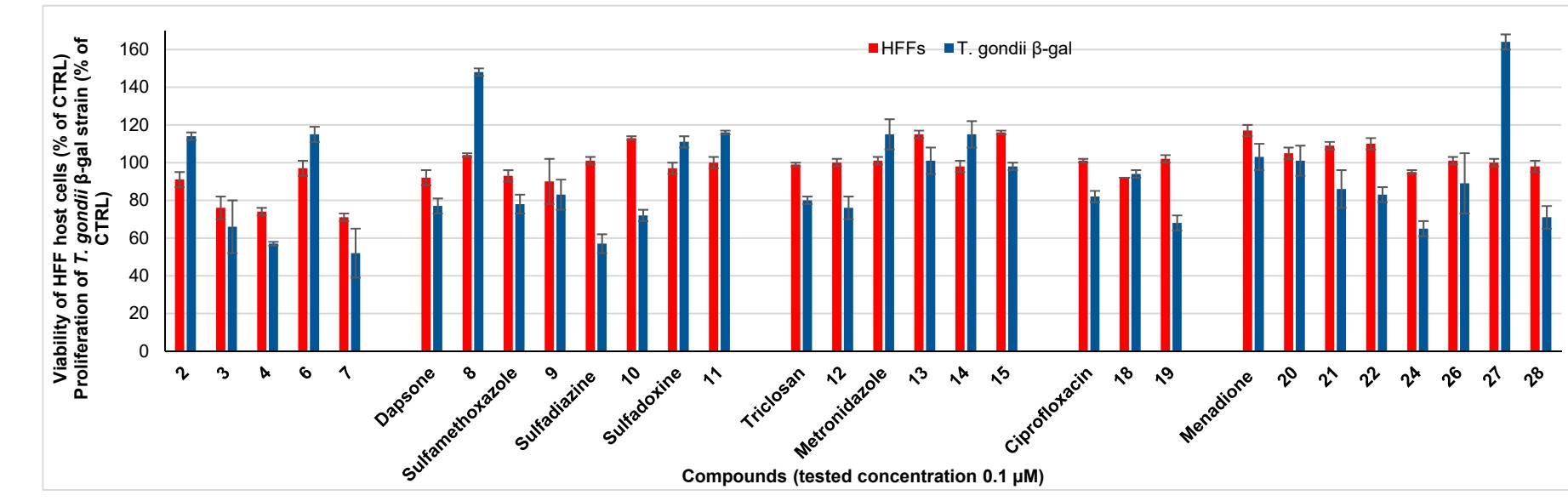

**Figure 6.** *Cont.*

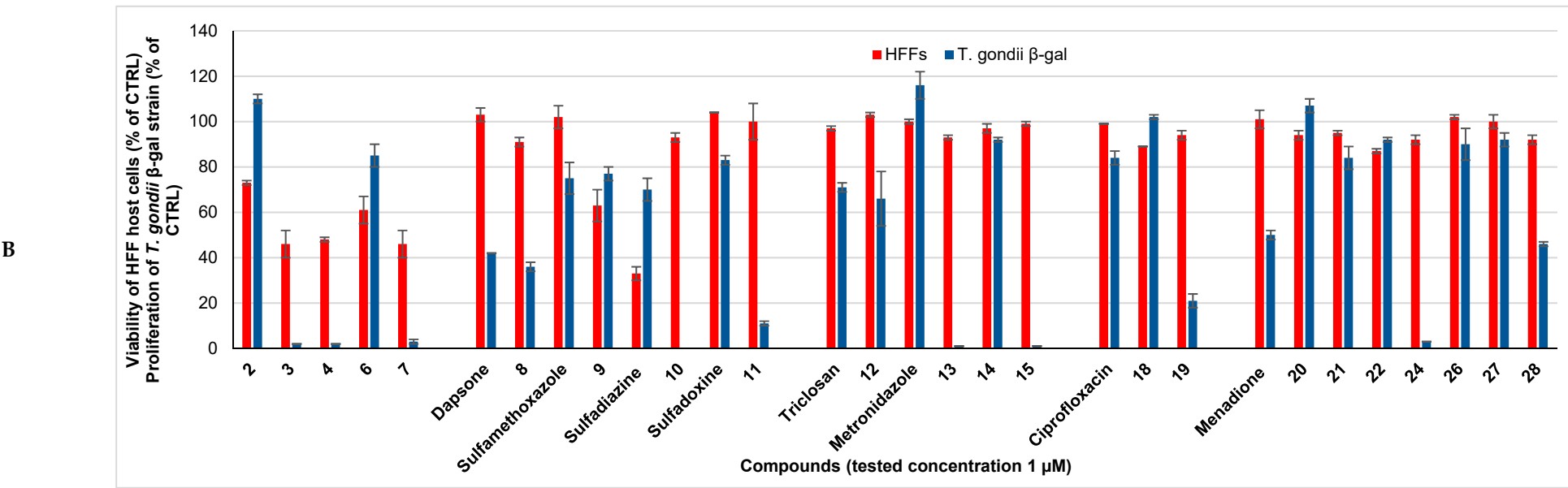

**Figure 6.** Clustered column chart showing the in vitro activities at 0.1 (**A**) and 1 μM (**B**) of the 30 tested compounds on HFF viability and *T. gondii* β-gal proliferation. Noninfected HFF monolayers treated only with 0.1% DMSO exhibited 100% viability and 100% proliferation was attributed to *T. gondii* β-gal tachyzoites treated only with 0.1% DMSO. Red bars represent viability values of HFF, and blue bars represent the proliferation of *T. gondii* β-gal tachyzoites. For each assay, standard deviations were calculated from triplicates and are displayed on the graph. Data for compounds **2–4** and **6** were previously reported in [30,43,80]. Complex **5** exhibited no activity against the parasite [30] (values not shown).

In a primary screening, transgenic *T. gondii* tachyzoites constitutively expressing β-galactosidase (*T. gondii* β-gal) were cultured in HFF monolayers and exposed to concentrations of 1 and 0.1 μM of each compound of interest. In parallel, the cytotoxicity of these compounds was evaluated at the same concentrations in noninfected HFF. As a measure of the parasite proliferation, β-galactosidase activity was determined, while the impact on noninfected HFF was assessed using the alamarBlue assay; the results are summarized in Table 4 and Figure 6.

Diruthenium Intermediates

From the diruthenium intermediates, carboxy and tri-amino derivatives **2** and **6** had no effect on parasite proliferation, but slightly affected HFF viability at 1 μM. In contrast, hydroxy, amino and alkyne functionalized compounds **3**, **4** and **7** drastically reduced *T. gondii* β-gal proliferation when administered at 1 μM but were also toxic to HFF already at 0.1 μM.

Antimicrobial Drugs and Conjugates

Except for sulfadiazine and sulfamethoxazole conjugate **9**, all antimicrobial drugs, intermediates based on the antimicrobials, and conjugates did not affect the viability of the HFF even at the highest tested concentration (1 μM) (Table 4, Figure 6).

The poor in vitro anti-*Toxoplasma* activity of the selected antimicrobial drugs agrees with some previously reported data [44].

If dapsone impaired parasite proliferation even at 0.1 μM, its conjugate **8** inhibited *T. gondii* β-gal proliferation to 36% only when applied at 1 μM. Both sulfamethoxazole and its conjugate **9** exhibited only reduced effect on the parasite. However, while sulfamethoxazole had little influence on HFF viability, **9** displayed considerable cytotoxicity to host cells at 1 μM.

Remarkably, the sulfadiazine conjugate **10** was not toxic to HFF but inhibited the *T. gondii* β-gal proliferation to 72% when administrated at 0.1 μM, and completely abolished it at 1 μM. Sulfadiazine was toxic to HFF when administrated at 1 μM, but at 0.1 μM, it did not affect the HFF viability but reduced the *T. gondii* β-gal proliferation to 57%. When applied at 1 μM, sulfadoxine-conjugate **11** exhibited a stronger effect on the *T. gondii* β-gal proliferation compared to sulfadoxine (11 vs. 83%).

All sulfa-drugs were connected to the trithiolato diruthenium unit via similar strong carbamide bonds. With the exception of the sulfamethoxazole conjugate **9**, the nature of the anchored organic drug had little effect on the viability of the host cells, but in some cases, impacted the efficacy against the parasite (e.g., sulfadiazine vs. **10** and sulfadoxine vs. **11**). Compared to the trithiolato diruthenium amino analogue **4**, conjugates with sulfa-drugs were all less cytotoxic to HFF.

Compared to the diruthenium carboxy intermediate **2**, ester conjugates with triclosan **12** and metronidazole **13** reduced parasite proliferation more efficiently at 1 μM.

Both triclosan and its ester conjugate **12** presented a similar reduced antiparasitic effect (at 1 μM parasite proliferation was reduced to 71 and 66%, respectively).

Although metronidazole and azide intermediate **14** displayed only low activity against *T. gondii* β-gal, conjugates **13** (ester) and **15** (triazole) were highly active at 1 μM and almost abolished proliferation. In this case, conjugation to the trithiolato diruthenium unit improved the antiparasitic activity without increasing host toxicity. Alkyne intermediate **7** was more active on the parasite than the carboxy precursor **2**, but also more toxic to HFF, while 'click' metronidazole conjugate **15** impacted less the HFF viability than **7**.

Compared to amino diruthenium compound **4**, amide conjugates with ciprofloxacin **18** and **19**, and with menadione **24** were less detrimental to host cell viability.

*N*-Boc protected ciprofloxacin conjugate **18** exhibited no antiparasitic activity, while the deprotected conjugate **19** had a stronger impact on *T. gondii* β-gal proliferation in comparison to ciprofloxacin (21 vs. 84% at 1 μM).

When administered at 1 μM, menadione reduced *T. gondii* β-gal proliferation to 50%, while its derivatives **20**, **21** and **22**, presenting different chain lengths between the 1,4-naphtoquinone unit and the carboxy group, exhibited little antiparasitic effect. Compared to menadione intermediate **20**, the amide conjugate **24** presented increased anti-*T. gondii* β-gal activity, almost abolishing parasite proliferation at 1 μM. Conjugates **26** and **27** presenting three menadione units connected via amide bonds to the trithiolato diruthenium core did not affect parasite proliferation, regardless of the linker size. A similar lack of antiparasitic effect has been previously observed in the case of a coumarin trisubstituted ester derivative [43] and might be associated with the important size of this type of analogue. This observation is corroborated by the results obtained for the monosubstituted conjugate **28**, which presented increased efficacy on the parasite compared to intermediate **22**. The difference in antiparasitic activity between the monosubstituted derivatives **24** and **28** might be due to the nature of the other two bridging thiols and the length of the linker between the diruthenium scaffold and the menadione moiety.

The first screening allowed the identification of five conjugates, **10** (sulfadiazine), **11** (sulfadoxine), **13** and **15** (metronidazole), **19** (ciprofloxacin) and **24** (menadione) that were more active against *T. gondii* at 1 μM compared to the respective antimicrobial drug. Concomitantly, these derivatives exhibited low or intermediate impairment of HFF viability at the highest tested concentration (1 μM). The highest antiparasitic activity increase was observed in the case of the metronidazole ester and triazole conjugates **13** and **15**, which almost abolished parasite proliferation when administered at 1 μM while the antimicrobial drug was inactive at the same concentration. The most modest amelioration was observed in the case of menadione and its respective conjugate **24** which, at 1 μM, reduced parasite proliferation with 50 and 3%, respectively.

### 2.2.2. Secondary Screening

Based on the primary screening, conjugates **10**, **11**, **13**, **15** and **24** were selected for the determination of the $IC_{50}$ values against *T. gondii* and the assessment of HFF viability after exposure to 2.5 μM. For the selection of the compounds for $IC_{50}$ determination, two criteria had to be simultaneously satisfied: (i) *T. gondii* β-gal growth inhibition of 90% or more compared to an untreated control when the compound was applied at 1 μM, and (ii) HFF host cell viability not impaired by more than 50% for a compound applied at 1 μM. The results are summarized in Table 5, and dose-response curves are shown in Figure S1 (Supplementary Materials).

**Table 5.** Half-maximal inhibitory concentration ($IC_{50}$) values (μM) on *T. gondii* β-gal for seven selected compounds and pyrimethamine (used as standard), and their effect at 2.5 μM on HFF viability.

| Compound | *T. gondii* β-gal $IC_{50}$ (μM) | [LS; LI] [b] | SE [c] | HFF Viability at 2.5 μM (%) [d] | SD [e] |
|---|---|---|---|---|---|
| **Pyrimethamine** | 0.326 | [0.396; 0.288] | 0.052 | 99 | 6 |
| *Ruthenium intermediates* | | | | | |
| **2** [a] | 0.181 | [1.482; 0.274] | 0.954 | 99 | 2 |
| **4** [a] | 0.153 | [0.185; 0.127] | 0.049 | 51 | 5 |
| *Conjugates with sulfa-drugs* | | | | | |
| **10** | 0.524 | [0.562; 0.488] | 0.069 | 62 | 1 |
| **11** | 0.063 | [0.072; 0.055] | 0.136 | 83 | 0 |
| *Conjugates with triclosan and metronidazole* | | | | | |
| **13** | 0.152 | [0.181; 0.127] | 0.175 | 64 | 3 |
| **15** | 0.500 | [0.884; 0.284] | 0.568 | 102 | 2 |
| *Conjugates with menadione* | | | | | |
| **24** | 0.481 | [0.525; 0.441] | 0.086 | 32 | 3 |

[a] Data for pyrimethamine, **2** and **4** were previously reported [30,43,80]. [b] Values at 95% confidence interval (CI); LS is the superior limit of CI and LI is the inferior limit of CI. [c] The standard error of the regression (SE), represents the average distance that the observed values fall from the regression line. [d] Control HFF cells treated only with 0.25% DMSO exhibited 100% viability. [e] The standard deviation of the mean (six replicate experiments).

For comparison, the results obtained for the carboxy and amino-functionalized diruthenium intermediates **2** and **4**, as well as those for pyrimethamine used as the standard, are also shown.

The most interesting compound of the series is the sulfadoxine conjugate **11**, exhibiting a low $IC_{50}$ (0.063 μM), and only slightly affecting the HFF viability at 2.5 μM (83%). Of note, the $IC_{50}$ value of **11** is significantly lower than that of the corresponding carboxy diruthenium intermediate (0.181 μM) or that of pyrimethamine (0.326 μM). Interestingly, the sulfadiazine derivative **10** presented poor antiparasitic activity ($IC_{50}$ 0.524 μM, more than 8 times higher compared to **11**), and increased toxicity to HFF (62%). These significant differences between these two conjugates underline the importance of the drug fragment for the biological activity, as both conjugates share the same diruthenium moiety and similar bonding between the two units. Sulfadoxine is interesting as this sulfonamide is used in combination with pyrimethamine in the treatment or prevention of malaria [98–101].

A significant difference was observed between the two metronidazole conjugates **13** and **15**. In comparison to **2**, the ester conjugate **13** presents a lower $IC_{50}$ value (0.152 vs. 0.181 μM), but increased HFF toxicity when administered at 2.5 μM (64 vs. 99%). The triazole conjugate **15** exhibited no HFF cytotoxicity but also reduced antiparasitic activity. As both compounds were obtained from the same carboxy diruthenium analogue **2**, the differences are likely due to the linking units between the diruthenium moiety and metronidazole.

The amide menadione hybrid **24** was not only more toxic to HFF (viability at 2.5 μM, 32 vs. 51%) but also less active in parasite proliferation inhibition compared to the corresponding amino intermediate **4** (0.481 vs. 0.153 μM).

Overall, the results obtained for conjugates **10**, **11**, **13**, **15** and **24** indicate that this type of hybrid molecules, antimicrobial drug-thiolato-bridged dinuclear ruthenium(II)–arene complex, seems promising and that a fine-tuning of the biological activity can be achieved by a judicious choice of the drugs and connecting units.

The mechanism of action of these trithiolato-bridged dinuclear ruthenium(II)–arene complexes and conjugates has not been yet elucidated. In contrast to almost all other ruthenium(II)–arene complexes presenting labile chlorine or carboxylate ligands, these dinuclear ruthenium(II)–arene compounds do not hydrolyze and are stable in the presence of DNA and amino acids [26]. Oxidation of cysteine (Cys) and glutathione (GSH) to form cystine and glutathione-disulfide (GSSG), respectively, was observed in the presence of some complexes. Still, no correlation between in vitro cytotoxicity and the catalytic activity on the oxidation reaction of glutathione could be established [24,102].

For some compounds, transmission electron microscopy (TEM) detected ultrastructural alterations in the matrix of the *T. gondii* mitochondria within few hours of treatment, followed by a more pronounced destruction of tachyzoites at later time points [30,43]. Gaining more insight into the mechanisms of action of these dinuclear complexes, responsible for the observed effects on various parasites, will allow a more rational selection of drugs that could be anchored to the diruthenium scaffold (e.g., organic molecules sharing the same molecular target or that can direct the diruthenium fragment to reach a specific biomolecule or organelle).

## 3. Experimental

### 3.1. Chemistry

The experimental chemistry portion, with a full description of experimental procedures and characterization data for all compounds, is presented in the Supplementary Materials.

### 3.2. Crystal-Structure Determination

A crystal of **9** ($C_{60}H_{74}ClN_3O_4Ru_2S_4 \cdot 3CH_3CH_2OH \cdot CHCl_3$) was mounted in the air at ambient conditions. All measurements were made on a *RIGAKU Synergy S* area-detector diffractometer [103] using mirror optics monochromated Cu *Kα* radiation (λ = 1.54184 Å) [104]. The unit cell constants and an orientation matrix for data collection were obtained from a least-squares refinement of the setting angles of reflections in the range $2.82° < θ < 77.133°$.

A total of 2404 frames were collected using ω scans, with 0.25 s exposure time, a rotation angle of 0.5° per frame, a crystal-detector distance of 65.0 mm, at T = 110(2) K.

Data reduction was performed using the *CrysAlisPro* [103] program. The intensities were corrected for Lorentz and polarization effects, and an absorption correction based on the multiscan method using SCALE3 ABSPACK in *CrysAlisPro* [103] was applied. Data collection and refinement parameters are given in Table 1.

The structure was solved by direct methods using *SHELXT* [105], which revealed the positions of all non-hydrogen atoms of the title compound. All non-hydrogen atoms were refined anisotropically. H-atoms were assigned in geometrically calculated positions and refined using a riding model where each H-atom was assigned a fixed isotropic displacement parameter with a value equal to 1.2 Ueq of its parent atom (1.5 Ueq for methyl groups).

The refinement of the structure was carried out on $F^2$ using full-matrix least-squares procedures, which minimized the function $\Sigma w(F_o{}^2 - F_c{}^2)^2$. The weighting scheme was based on counting statistics and included a factor to down-weight the intense reflections. All calculations were performed using the *SHELXL-2014/7* [106] program in OLEX2 [107].

### 3.3. In Vitro Activity Assessment against T. gondii Tachyzoites and HFF

All tissue culture media were purchased from Gibco-BRL, and biochemical agents from Sigma-Aldrich. Human foreskin fibroblasts (HFF) were purchased from ATCC, maintained in DMEM (Dulbecco's Modified Eagle's Medium) supplemented with 10% fetal calf serum (FCS, Gibco-BRL, Waltham, MA, USA) and antibiotics as previously described [108]. Transgenic *T. gondii* β-gal samples (expressing the β-galactosidase gene from *Escherichia coli*) were kindly provided by Prof. David Sibley (Washington University, St. Louis, MO, USA) and were maintained, isolated, and prepared for new infections as shown before [108,109].

All the compounds were prepared as 1 mM stock solutions from powder in dimethyl sulfoxide (DMSO, Sigma, St. Louis, MO, USA). For in vitro activity and cytotoxicity assays, HFF were seeded at $5 \times 10^3$/well and allowed to grow to confluence in phenol-red free culture medium at 37 °C and 5% $CO_2$. Transgenic *T. gondii* β-gal tachyzoites were isolated and prepared for infection as described [108]. *T. gondii* tachyzoites were released from host cells, and HFF monolayers were infected with freshly isolated parasites ($1 \times 10^3$/well), and compounds were added concomitantly with infection. In the primary screening, HFF monolayers infected with *T. gondii* β-gal received 0.1 and 1 μM of each compound, or the corresponding concentration of DMSO (0.01 or 0.1% respectively) as controls and incubated for 72 h at 37 °C/5% $CO_2$ as previously described [96].

For the next step, $IC_{50}$ measurements for *T. gondii* β-gal were performed. The selected compounds were added concomitantly with infection in 8 serial concentrations 0.007, 0.01, 0.03, 0.06, 0.12, 0.25, 0.5, and 1 μM. After a period of 72 h of culture at 37 °C/5% $CO_2$, the culture medium was aspirated, and cells were permeabilized by adding 90 μL PBS (phosphate-buffered saline) with 0.05% Triton X-100. After the addition of 10 μL 5 mM chlorophenolred-β-D-galactopyranoside (CPRG; Roche Diagnostics, Rotkreuz, Switzerland) in PBS, the absorption shift was measured at 570 nm wavelength at various time points using an EnSpire® multimode plate reader (PerkinElmer, Inc., Waltham, MA, USA).

For the primary screening at 0.1 and 1 μM, activity was measured as the release of chlorophenol red over time, was calculated as a percentage from the respective DMSO control, which represented 100% of *T. gondii* β-gal growth. For the $IC_{50}$ assays, the activity measured as the release of chlorophenol red over time was proportional to the number of live parasites down to 50 per well as determined in pilot assays. $IC_{50}$ values were calculated after the logit-log-transformation of relative growth and subsequent regression analysis.

All calculations were performed using the corresponding software tool contained in the Excel software package (Microsoft, Redmond, WA, USA). Cytotoxicity assays using uninfected confluent HFF host cells were performed by the alamarBlue assay as previously reported [110]. Confluent HFF monolayers in 96 well-plates were exposed to 0.1, 1 and

2.5 μM of each compound. Non-treated HFF as well as DMSO controls (0.01%, 0.1% and 0.25%) were included. After 72 h of incubation at 37 °C/5% $CO_2$, the medium was removed, and plates were washed once with PBS. 200 μL of Resazurin (1:200 dilution in PBS) were added to each well. Plates were measured at excitation wavelength 530 nm and emission wavelength 590 nM at the EnSpire® multimode plate reader (PerkinElmer, Inc.). Fluorescence was measured at different time points. Relative fluorescence units were calculated from time points with linear increases.

## 4. Conclusions

This study has focused on the synthesis and in vitro evaluation of 13 new conjugates based on trithiolato-bridged ruthenium(II)–arene scaffold tethered with various antimicrobial drugs, aiming at improving the antiparasitic properties and the selectivity.

The type of chemical bond between the two units and their relative proportion was varied. In total, 30 compounds (conjugates, representative intermediates, drugs) were submitted to a first activity screening against *T. gondii* β-gal tachyzoites cultured in HFF and cytotoxicity determination against HFF host cells, which allowed the identification of five interesting conjugates. The $IC_{50}$ values against *T. gondii* and the evaluation of HFF viability after exposure to 2.5 μM led to the selection of the sulfadoxine conjugate **11** as the most promising of this series of 13 conjugates.

Our study suggests that the nature of the drug and of the linker between the drug and the diruthenium(II) moiety greatly impacts biological activity. Overall, anchoring antimicrobial drugs to trithiolato diruthenium(II)–arene moieties is a promising approach for obtaining new compounds presenting different toxicity profiles than the parent organometallic complexes. The conjugates obtained in this study deserve further attention and can be evaluated for other pharmacological applications (e.g., antiproliferative activity on cancer cells or as antibacterials).

**Supplementary Materials:** The Supplementary Materials are available online at https://www.mdpi.com/article/10.3390/inorganics9080059/s1. The dose response curves for compounds **10**, **11**, **13**, **15** and **24** as inhibitors of *T. gondii* β-gal tachyzoites proliferation and the experimental chemistry portion with the full description of experimental procedures. Accession code CCDC 2084579 (compound **9**) contains the supplementary crystallographic data for this paper. These data can be obtained free of charge via www.ccdc.cam.ac.uk/data_request/cif, or by emailing data_request@ccdc.cam.ac.uk, or by contacting The Cambridge Crystallographic Data Centre, 12 Union Road, Cambridge CB21EZ, UK; fax: +44 1223 336033.

**Author Contributions:** Conceptualization, E.P., J.F. and A.H.; methodology, E.P., O.D.; J.F., N.A. and A.H.; software, O.D., S.K.J., E.P., G.B., N.A. and G.B.; validation, E.P., J.F., G.B. and N.A.; formal analysis, O.D., S.K.J., E.P., N.A., G.B. and A.H.; investigation, O.D., S.K.J., E.P., J.F., G.B., N.A. and A.H.; resources, J.F. and A.H.; data curation, E.P., O.D., S.K.J., J.F., N.A., G.B. and A.H.; writing-original draft preparation, E.P., J.F., N.A.; writing-review and editing, E.P., J.F., O.D., N.A., A.H.; visualization, E.P., O.D., J.F., N.A.; supervision, E.P., O.D., J.F. and A.H.; project administration, J.F. and A.H.; funding acquisition, J.F. and A.H. All authors have read and agreed to the published version of the manuscript.

**Funding:** The authors declare no competing financial interest. This work was financially supported by the Swiss Science National Foundation (SNF, Sinergia project CRSII5-173718 and project 310030_184662) (E.P., O.D., J.F., A.H., G.B., N.A.). S.K.J. was supported by an ERASMUS grant and by a Swiss Confederation mobility grant. Y.A. was supported by a Swiss Governmental Excellence Fellowship. The Synergy diffractometer was partially funded by the SNF within the R'Equip program (project 206021_177033).

**Institutional Review Board Statement:** Not applicable.

**Informed Consent Statement:** Not applicable.

**Acknowledgments:** The X-ray crystal structure determination service unit of the Department of Chemistry, Biochemistry and Pharmaceutical Sciences of the University of Bern is acknowledged for measuring, solving, refining, and summarizing the structure of compound **9**.

**Conflicts of Interest:** The authors declare no conflict of interest.

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
