# Peer review of "Synthesis and Antiparasitic Activity of New Conjugates—Organic Drugs Tethered to Trithiolato-Bridged Dinuclear Ruthenium(II)–Arene Complexes"

_inorganics, doi:10.3390/inorganics9080059_

Round 1

Reviewer 1 Report

The manuscript evaluates 30 new di-ruthenium(II)arene compounds linked to anti-microbial compounds to be employed as anti-parasitic agents. Their cytotoxicity was determined, and some exhibited low IC50 values. The new ruthenium complexes were synthesized through modification of the corresponding thiolate-bridged ruthenium dimers. The new complexes were characterized by NMR, ESI-MS and some by X-ray diffraction. The new complexes were investigated to assess their activity against T. gondii tachyzoites and human foreskin fibroblasts host cells.

Overall, the study touches the important topic of organometallic compounds as anti-bacterial agents. This is a worthwhile pursuit. The new organometallic drugs do not really outperform known drugs, but perform comparable to those mentioned by the authors. There seems some structure-activity relationship, but it is not very pronounced. While not groundbreaking, the results provide a few datapoints in the realm of “organometallic drugs”. As such, the manuscript deserves publication.

The authors may consider the following points.

The authors may want to add some assessment of the stability of the ruthenium complexes, especially under physiological conditions. If the complexes decompose easily, the decomposition products may be the actual active ingredients, which should be excluded.

I mentioned above that the structure-activity relationships observed by the authors are weak. A comment to that effect may help the reader understand the significance. However, as the ruthenium complexes perform comparably, some of them may also have a common decomposition product which may be pharmaceutically active.

Some technical points. Schemes 1 to 8 might be easier to read if the authors add isolated yields to the products. In the calculation of elemental analyses in the supporting information, in some cases methanol solvent is factored in. Can the residual solvent also be seen in the NMR spectra? I think factoring in residual solvent is only appropriate if the presence of the solvent is established by a different, spectroscopic method.

Triclosan (5-chloro-2-(2,4-dichlorophenoxy)phenol, Figure 2), should “Figure 2” be “Figure 1”?

Author Response

Referee 1

The manuscript evaluates 30 new di-ruthenium(II)arene compounds linked to anti-microbial compounds to be employed as anti-parasitic agents. Their cytotoxicity was determined, and some exhibited low IC50 values. The new ruthenium complexes were synthesized through modification of the corresponding thiolate-bridged ruthenium dimers. The new complexes were characterized by NMR, ESI-MS and some by X-ray diffraction. The new complexes were investigated to assess their activity against T. gondii tachyzoites and human foreskin fibroblasts host cells.

Overall, the study touches the important topic of organometallic compounds as anti-bacterial agents. This is a worthwhile pursuit. The new organometallic drugs do not really outperform known drugs, but perform comparable to those mentioned by the authors. There seems some structure-activity relationship, but it is not very pronounced. While not groundbreaking, the results provide a few datapoints in the realm of “organometallic drugs”. As such, the manuscript deserves publication.

General answer: We thank the reviewer for his/her good comments and useful and constructive observations and advices.

The authors may consider the following points.

The authors may want to add some assessment of the stability of the ruthenium complexes, especially under physiological conditions. If the complexes decompose easily, the decomposition products may be the actual active ingredients, which should be excluded.

I mentioned above that the structure-activity relationships observed by the authors are weak. A comment to that effect may help the reader understand the significance. However, as the ruthenium complexes perform comparably, some of them may also have a common decomposition product which may be pharmaceutically active.

Answer:

The reduced solubility of the compounds in solutions mimicking physiological solutions in concentration ranges suitable for NMR made experiments to assess compounds stability less straightforward. However, we know for long time that these complexes are stable in complexing solvents like DMSO and acetonitrile, even after long storage periods as proven by 1H-NMR experiments. F. Giannini et al. Inorg. Chem.  50, 10552-10554, 2011, F. Giannini et al. J. Biol. Inorg. Chem. 17, 951-960, 2012, J. Furrer , G. Süss-Fink Coord. Chem. Rev. 309, 36-50, 2016, O. Desiatkina et al. ChemBioChem 21, 2818-2835, 2020, V. Studer et al, Pharmaceuticals 13, 471, 2020. E. Păunescu et al. Eur J Med Chem 2021.)

A small explicative note was included in the manuscript at the end of paragraph 2.1.5

For the assessment of the biological activity, the compounds were prepared as stock solutions in dimethylsulfoxide (DMSO), in which the compounds are well soluble. 1H-NMR spectra of similar conjugates (polypeptides, coumarin units or conjugates with two or three diruthenium units) dissolved in DMSO-d6 or deuterated water, recorded at 25°C 5 min and 28 days after sample preparation showed no visible changes, demonstrating a very good stability of the conjugates in this highly complexing solvent and in water. [42,43, 89,117]

Some technical points.

Schemes 1 to 8 might be easier to read if the authors add isolated yields to the products.

Answer:

We only found very few manuscripts in Inorganics that has implemented this way of reporting the yields. However, we have followed the recommendation of this referee and have adapted the schemes accordingly.

In the calculation of elemental analyses in the supporting information, in some cases methanol solvent is factored in. Can the residual solvent also be seen in the NMR spectra? I think factoring in residual solvent is only appropriate if the presence of the solvent is established by a different, spectroscopic method.

Answer:

As visible in the various NMR spectra, probably also due to their 'salt' nature, this type of compounds very often include/precipitate with H2O or molecules of solvents used for their purification, and which are difficult to eliminate even by drying the compounds for 72 h under high vacuum.

Triclosan (5-chloro-2-(2,4-dichlorophenoxy)phenol, Figure 2), should “Figure 2” be “Figure 1”?

Answer:

We thank the reviewer for pointing this mistake. We have corrected the text accordingly:

''Triclosan (5-chloro-2-(2,4-dichlorophenoxy)phenol, Figure 1) is an antibacterial which was shown to inhibit the in vitro proliferation of T. gondii tachyzoites in the low nanomolar range.''

Reviewer 2 Report

The manuscript presents an experimental study of a large set of compounds composed of a di-ruthenium(II)arene unit linked with various antimicrobial drugs. The subject of the study is interesting and the motivation for this study is clear. However, I am not sure whether this study falls into the scope of interest of the journal. The manuscript focused mainly on the biological activity of the aforementioned compounds and the inorganic aspect of this study is restricted to the synthesis of these compounds and to the determination of the structure of one selected compound. Therefore, I think that the manuscript is appropriate to a biochemistry/bioinorganic chemistry-oriented journal rather than a typical inorganic chemistry journal, such as Inorganics. In my opinion, the Authors should study the structure of the compounds in a greater detail if the manuscript has to be published in Inorganics. I recommend the Authors should consider a biochemistry-oriented MDPI journal instead.

Other remarks:
1. The manuscript is rather long and some of its parts do not read well. For example, the introduction is too long and reading this section is tedious. On page 2 there is a paragraph describing toxoplasmosis -- I am not sure that this paragraph is interesting to inorganic chemists in general. The introduction quotes about 90 references -- I think that the Authors could prepare a separate short review devoted to the subject instead of inserting such a detailed introduction into a research paper. Section 2.2 reminds me of a part of a report (many very short paragraphs without in-depth discussion) rather than a brief presentation of results.

2. In section 2.1.6 the Authors state: "To the best of our knowledge, this is the first example of a structure containing the trithiolato bridged di-ruthenium moiety and an organic moiety." In my opinion, this point is most interesting to inorganic chemists and it should be discussed in a greater detail. On page 13, there is a single short paragraph discussing the structure of compound 9 and the structure 9 is still far away from being fully understood. The analysis of intermolecular contacts is carried out using only geometrical parameters -- I advise the Authors to perform the Bader analysis of the electron charge and the analysis of Hirshfeld surface if they want to examine the structure in a greater detail. Obviously, there are tons of computational chemistry methods that can provide many valuable insights in the energetic aspect of the structure of the compounds and in the bonding pattern of the compounds. Besides, it is not clear whether hydrogen bonding interactions are the only intermolecular interactions in the crystal of compound 9. Additional characterization of intramolecular hydrogen bonding interactions (Table 3) would also be beneficial for improving the quality of section 2.1.6.

Minor improvements I can suggest:
1. Figure 1: Please, explain in the caption to what compounds the R groups shown in the upper right corner should be inserted.
2. Table 3: the numbering of atoms seems rather mysterious. A figure presenting the numbering of atoms in the molecule of compound 9 is necessary.  

Author Response

Referee 2

The manuscript presents an experimental study of a large set of compounds composed of a di-ruthenium(II)arene unit linked with various antimicrobial drugs. The subject of the study is interesting and the motivation for this study is clear. However, I am not sure whether this study falls into the scope of interest of the journal. The manuscript focused mainly on the biological activity of the aforementioned compounds and the inorganic aspect of this study is restricted to the synthesis of these compounds and to the determination of the structure of one selected compound. Therefore, I think that the manuscript is appropriate to a biochemistry/bioinorganic chemistry-oriented journal rather than a typical inorganic chemistry journal, such as Inorganics. In my opinion, the Authors should study the structure of the compounds in a greater detail if the manuscript has to be published in Inorganics. I recommend the Authors should consider a biochemistry-oriented MDPI journal instead.

Answer:

The manuscript should be published in the organometallics section of Inorganics, in a special edition dedicated to metal-arene complexes. Bioinorganic chemistry, metal-based drugs belong to the keywords of this issue, we therefore feel that our manuscript fits well in Inorganics.

For more than one decade, the investigation of the biological properties (anticancer, antimicrobial) represents the main application of these arene ruthenium compounds.

The compounds which were submitted to the biological activity assessment were analysed and characterised by, 1H, 13C, 19F (where suited), ESI-MS and elemental analysis (data presented in the Supporting information). Certainly, in further studies, other analyses can be considered (e.g. IR, UV-Vis), for compounds that present interesting biological properties (high activity and selectivity).

Other remarks:

  1. The manuscript is rather long and some of its parts do not read well. For example, the introduction is too long and reading this section is tedious. On page 2 there is a paragraph describing toxoplasmosis -- I am not sure that this paragraph is interesting to inorganic chemists in general. The introduction quotes about 90 references -- I think that the Authors could prepare a separate short review devoted to the subject instead of inserting such a detailed introduction into a research paper.

Answer:

The paragraph resuming some information about the Toxoplasmosis is indeed not of great interest for inorganic chemists, but it is essential for framing the subject and introduce the field in which this type of compounds could find applications. Bioinorganic chemists will probably find this part more interesting.

Section 2.2 reminds me of a part of a report (many very short paragraphs without in-depth discussion) rather than a brief presentation of results.

Answer:

In Section 2.2 we sought to compare and discuss the biological activity of various compounds based on their structure. We have profoundly modified this section and tried to shorten it to a certain extent. We hope it is now clearer and more comprehensible.

  1. In section 2.1.6 the Authors state: "To the best of our knowledge, this is the first example of a structure containing the trithiolato bridged di-ruthenium moiety and an organic moiety." In my opinion, this point is most interesting to inorganic chemists and it should be discussed in a greater detail. On page 13, there is a single short paragraph discussing the structure of compound 9 and the structure 9 is still far away from being fully understood. The analysis of intermolecular contacts is carried out using only geometrical parameters -- I advise the Authors to perform the Bader analysis of the electron charge and the analysis of Hirshfeld surface if they want to examine the structure in a greater detail. Obviously, there are tons of computational chemistry methods that can provide many valuable insights in the energetic aspect of the structure of the compounds and in the bonding pattern of the compounds. Besides, it is not clear whether hydrogen bonding interactions are the only intermolecular interactions in the crystal of compound 9. Additional characterization of intramolecular hydrogen bonding interactions (Table 3) would also be beneficial for improving the quality of section 2.1.6.

Answer:

To the best of our knowledge there are so far only three other reports which describe conjugates of trithiolato dinuclear ruthenium(II)-arene with various types of organic compounds anchored (i.e. chlorambucil, polypeptides, coumarin units, all of them are mentioned in the text of the manuscript). However, none of these previous studies reports a single-crystal X-ray structure. The results (the structure presented in Figure 3) confirm the expected structure. Considering the aim of this study and that compound 9 in not interesting for further development (at least not as antiparasitic drug against Toxoplasmosis) we do not consider necessary at this point to do extended calculations on the structure on this compound. Figure 5 has been corrected presented the labels of the atoms involved in the H-bonding interactions for which the parameters are presented in Table 3. The data presented in Table 3 (bond lengths and angles) correspond to previous reported values that characterise this type of interactions.

The organic molecule anchored on the di-ruthenium unit presents H-bond acceptors and donors and this favour the organization in dimers in network. Certainly, there are not the only interactions present in the X-ray structure of this compound as it can be seen in Figure 5 (corrected) the solvent molecules (EtOH) trapped in the network. We presented, as example, the H-bonding interactions to which the conjugate participates. An exhaustive study of the structure of compound not intended as it is not in the purpose of this study.

Minor improvements I can suggest:
1. Figure 1: Please, explain in the caption to what compounds the R groups shown in the upper right corner should be inserted.

Answer:

The R group for the sulfa-drugs presented in Figure 1. was replaced with R'.

  1. Table 3: the numbering of atoms seems rather mysterious. A figure presenting the numbering of atoms in the molecule of compound 9 is necessary.  

Answer:

The structure presented in Figure 5 was corrected to include the numbering on the atoms involved in the H-bonding interactions described in Table 3.

Round 2

Reviewer 2 Report

Referee 2 comment:
The manuscript presents an experimental study of a large set of compounds composed of a di-ruthenium(II)arene unit linked with various antimicrobial drugs. The subject of the study is interesting and the motivation for this study is clear. However, I am not sure whether this study falls into the scope of interest of the journal. The manuscript focused mainly on the biological activity of the aforementioned compounds and the inorganic aspect of this study is restricted to the synthesis of these compounds and to the determination of the structure of one selected compound. Therefore, I think that the manuscript is appropriate to a biochemistry/bioinorganic chemistry-oriented journal rather than a typical inorganic chemistry journal, such as Inorganics. In my opinion, the Authors should study the structure of the compounds in a greater detail if the manuscript has to be published in Inorganics. I recommend the Authors should consider a biochemistry-oriented MDPI journal instead.

Authors' answer:
The manuscript should be published in the organometallics section of Inorganics, in a special edition dedicated to metal-arene complexes. Bioinorganic chemistry, metal-based drugs belong to the keywords of this issue, we therefore feel that our manuscript fits well in Inorganics.
For more than one decade, the investigation of the biological properties (anticancer, antimicrobial) represents the main application of these arene ruthenium compounds. 
The compounds which were submitted to the biological activity assessment were analysed and characterised by, 1H, 13C, 19F (where suited), ESI-MS and elemental analysis (data presented in the Supporting information). Certainly, in further studies, other analyses can be considered (e.g. IR, UV-Vis), for compounds that present interesting biological properties (high activity and selectivity).

Referee 2 reply:
If the editor of the special issue thinks that this manuscript falls into the scope of this issue, I can understand that.

----------------------------------------

Referee 2 comment:
The manuscript is rather long and some of its parts do not read well. For example, the introduction is too long and reading this section is tedious. On page 2 there is a paragraph describing toxoplasmosis -- I am not sure that this paragraph is interesting to inorganic chemists in general. The introduction quotes about 90 references -- I think that the Authors could prepare a separate short review devoted to the subject instead of inserting such a detailed introduction into a research paper.

Authors' answer:
The paragraph resuming some information about the Toxoplasmosis is indeed not of great interest for inorganic chemists, but it is essential for framing the subject and introduce the field in which this type of compounds could find applications. Bioinorganic chemists will probably find this part more interesting.

Referee 2 reply:
I still think that the introduction is far too long for a research paper and the excessive length of the introduction will probably discourage many readers from reading the following parts of the manuscript.

-------------------------------------------

Referee 2 comment:
Section 2.2 reminds me of a part of a report (many very short paragraphs without in-depth discussion) rather than a brief presentation of results.

Authors' answer:
In Section 2.2 we sought to compare and discuss the biological activity of various compounds based on their structure. We have profoundly modified this section and tried to shorten it to a certain extent. We hope it is now clearer and more comprehensible.

Referee 2 reply:
OK

---------------------------------------------

Referee 2 comment:
In section 2.1.6 the Authors state: "To the best of our knowledge, this is the first example of a structure containing the trithiolato bridged di-ruthenium moiety and an organic moiety." In my opinion, this point is most interesting to inorganic chemists and it should be discussed in a greater detail. On page 13, there is a single short paragraph discussing the structure of compound 9 and the structure 9 is still far away from being fully understood. The analysis of intermolecular contacts is carried out using only geometrical parameters -- I advise the Authors to perform the Bader analysis of the electron charge and the analysis of Hirshfeld surface if they want to examine the structure in a greater detail. Obviously, there are tons of computational chemistry methods that can provide many valuable insights in the energetic aspect of the structure of the compounds and in the bonding pattern of the compounds. Besides, it is not clear whether hydrogen bonding interactions are the only intermolecular interactions in the crystal of compound 9. Additional characterization of intramolecular hydrogen bonding interactions (Table 3) would also be beneficial for improving the quality of section 2.1.6.

Authors' answer:
To the best of our knowledge there are so far only three other reports which describe conjugates of trithiolato dinuclear ruthenium(II)-arene with various types of organic compounds anchored (i.e. chlorambucil, polypeptides, coumarin units, all of them are mentioned in the text of the manuscript). However, none of these previous studies reports a single-crystal X-ray structure. The results (the structure presented in Figure 3) confirm the expected structure. Considering the aim of this study and that compound 9 in not interesting for further development (at least not as antiparasitic drug against Toxoplasmosis) we do not consider necessary at this point to do extended calculations on the structure on this compound. Figure 5 has been corrected presented the labels of the atoms involved in the H-bonding interactions for which the parameters are presented in Table 3. The data presented in Table 3 (bond lengths and angles) correspond to previous reported values that characterise this type of interactions.
The organic molecule anchored on the di-ruthenium unit presents H-bond acceptors and donors and this favour the organization in dimers in network. Certainly, there are not the only interactions present in the X-ray structure of this compound as it can be seen in Figure 5 (corrected) the solvent molecules (EtOH) trapped in the network. We presented, as example, the H-bonding interactions to which the conjugate participates. An exhaustive study of the structure of compound not intended as it is not in the purpose of this study.

Referee 2 reply:
As the Authors stated that compound 9 is not interesting for further development, section 2.1.6 is of no use to the study and it should be removed from the manuscript. This section in its current form is too superficial to present it in the manuscript. 

Author Response

Referee 2 reply:
I still think that the introduction is far too long for a research paper and the excessive length of the introduction will probably discourage many readers from reading the following parts of the manuscript.

Authors' answer:

The Introduction was shortened.

Referee 2 reply:
As the Authors stated that compound 9 is not interesting for further development, section 2.1.6 is of no use to the study and it should be removed from the manuscript. This section in its current form is too superficial to present it in the manuscript. 

Authors' answer:

The authors consider that manuscript section '2.1.6. X-ray crystallography' brings important and relevant information to the study.

Seen the results of the first screening presented in the manuscript, compound 9 is not interesting for further development as anti-toxoplasma agent. Nevertheless, this compound along with other conjugates reported in this manuscript are currently considered for other applications as anticancer or antibacterial drug.

As mentioned in the text of the manuscript, this single-crystal X-ray diffraction data confirm the expected structure of the sulfamethoxazole conjugate 9. Moreover, this constitutes the first example of a structure containing the trithiolato bridged diruthenium unit and an organic moiety, and brings originality to this study.

Table 1 presents the crystal and structure refinement data. Accession code CCDC 2084579 contain the supplementary crystallographic data for this structure. These data can be obtained free of charge via www.ccdc.cam.ac.uk/data_request/cif, or by emailing data_request@ccdc.cam.ac.uk, or by contacting The Cambridge Crystallographic Data Centre, 12 Union Road, Cambridge CB21EZ, UK; fax: +44 1223 336033.

Moreover, some network interactions are presented in Figure 6 and Table 3.

Recent examples of other X-ray crystal structures of trithiolato dinuclear trithiolato-bridged ruthenium(II)-arene complexes (not conjugates) are presented in the following publications:

Păunescu, E.; Boubaker, G.; Desiatkina, O.; Anghel, N.; Amdouni, Y.; Hemphill, A.; Furrer, J., The quest of the best – A SAR study of trithiolato-bridged dinuclear ruthenium(II)-arene compounds presenting antiparasitic properties. Eur. J. Med. Chem. 2021, 222, 113610. https://doi.org/10.1016/j.ejmech.2021.113610

Desiatkina, O.; Paunescu, E.; Mosching, M.; Anghel, N.; Boubaker, G.; Amdouni, Y.; Hemphill, A.; Furrer, J., Coumarin-tagged dinuclear trithiolato-bridged ruthenium(II)arene complexes: photophysical properties and antiparasitic activity. ChemBioChem 2020, 21 (19), 2818-2835. https://doi.org/10.1002/cbic.202000174.